# Three-Dimensional Reconstruction and Deformation Identification of Slope Models Based on Structured Light Method

**DOI:** 10.3390/s24030794

**Published:** 2024-01-25

**Authors:** Zhijian Chen, Changxing Zhang, Zhiyi Tang, Kun Fang, Wei Xu

**Affiliations:** 1Faculty of Civil Engineering and Mechanics, Kunming University of Science and Technology, Kunming 650500, China; chenzhijian@stu.kust.edu.cn (Z.C.); tang@kust.edu.cn (Z.T.); 2Intelligent Infrastructure Operation and Maintenance Technology Innovation Team, Yunnan Provincial Department of Education, Kunming University of Science and Technology, Kunming 650500, China; 3Faculty of Engineering, China University of Geosciences, Wuhan 430074, China; kunfang@cug.edu.cn

**Keywords:** structured light, physical model test, morphing analysis

## Abstract

In this study, we propose a meticulous method for the three-dimensional modeling of slope models using structured light, a swift and cost-effective technique. Our approach aims to enhance the understanding of slope behavior during landslides by capturing and analyzing surface deformations. The methodology involves the initial capture of images at various stages of landslides, followed by the application of the structured light method for precise three-dimensional reconstructions at each stage. The system’s low-cost nature and operational convenience make it accessible for widespread use. Subsequently, a comparative analysis is conducted to identify regions susceptible to severe landslide disasters, providing valuable insights for risk assessment. Our findings underscore the efficacy of this system in facilitating a qualitative analysis of landslide-prone areas, offering a swift and cost-efficient solution for the three-dimensional reconstruction of slope models.

## 1. Introduction

Landslides are typical natural disaster phenomena and have profoundly and consistently influenced people’s living. Landslides represent the most destructive phenomena, posing a significant threat to both human life and infrastructure. Therefore, the measurement and monitoring of landslide deformation behavior hold crucial significance in the prevention and mitigation of losses resulting from landslide disasters [1,2,3]. The physical slope model as an effective approach has been widely adopted to investigate the landslide mechanism.

Physical model testing is a prevalent method in the investigation of landslide disasters [4,5,6]. This approach effectively elucidates various mechanisms underlying landslide phenomena [7,8,9]. By constructing slope physical models in the laboratory, the types of slopes and disasters can be pre-designed for experimentation, aiming to achieve optimal simulation conditions [10,11]. Real-time monitoring of the slope model can be accomplished by installing various sensors and integrating them with software processing systems. The monitoring information obtained in the laboratory provides reliable experimental data for numerical simulation analysis. Therefore, based on the availability and effectiveness of physical model testing [12], extensive research has been conducted on slope system reinforcement and deformation behavior through customized slope physical models [13,14,15]. Pipatpongsa and Fang et al. [16] investigated the loading path and failure range of the slope model positioned on the cushion plane through slope model tests. Yin and Deng et al. [17] simulated the complete reverse slope failure process using a large inclined platform. Wang [18] investigated the behaviors of landslides reinforced with pile-anchors through slope model tests. From the information gathered during the monitoring of various landslides, the surface morphology and overall deformation of slopes constitute critical factors in slope monitoring.

The data obtained from existing slope model measurement systems can be categorized into one-dimensional, two-dimensional, and three-dimensional data. Common methods of obtaining one-dimensional data encompass the utilization of fiber optic grating sensors, inclinometers, array displacement transducers [19], and others. While sensors can provide relatively precise one-dimensional deformation data for slope models, an excess of sensors may not always be advantageous, as it can impact the structural integrity of the model and alter the deformation behavior of the slope [20]. Two-dimensional data are typically obtained through techniques such as Particle Image Velocimetry (PIV) [21,22] and Digital Image Correlation (DIC) [23,24]. In comparison to two-dimensional data, three-dimensional data for slope models allow for a more intuitive depiction of the slope’s failure process. Common non-contact methods for acquiring 3D data include the use of 3D laser scanners [25,26], binocular vision [27,28], photometric stereo [29,30], and structured light [31,32,33,34,35,36,37,38,39,40,41,42,43]. Three-dimensional laser scanners are expensive and time-consuming, limiting their application to small-scale physical slope model tests. The reconstruction area of binocular vision needs to be within the common field of view of the two cameras, posing a challenge to the specific operations involved in the slope model reconstruction process. Photometric stereo can be applied to objects of various sizes with high resolution, by adjusting the low-cost hardware. However, shiny or reflective surfaces may produce specular highlights, which could distort the images captured and result in inaccuracies in the reconstructed surface models. Though machine learning-based methods can improve this issue [44,45,46], the approach is too complex. Dealing with specular reflections often requires more complex algorithms and can limit the method’s applicability to a range of materials. In contrast, the structured light method offers a range of advantages in slope model reconstruction, including fast modeling speed, convenient system setup, and low cost.

Therefore, in this study, a structured-light-based method is utilized to capture three-dimensional point cloud data of the slope model. Using a gypsum sphere as a reference, the accuracy of the structured light system is validated. By adjusting the slope model to simulate slope landslide disasters, three-dimensional point cloud models of the slope at each stage of the landslide are generated. The deformation of the slope model is analyzed, and the critical areas where landslide disasters occur are identified.

## 2. Structured Light System

The structured light method is a three-dimensional imaging technique based on optical triangulation. As depicted in Figure 1, the structured light three-dimensional imaging system projects encoded structured light onto the surface of the object under examination using a projector. The camera captures the distorted pattern arising from the modulation of the object’s height, and the distorted pattern is subsequently demodulated using a computer. Ultimately, the three-dimensional shape of the object can be determined.

According to different structured light encoding strategies, the structured light field can be categorized into an intensity-modulated light field and a phase-modulated light field. Common intensity-modulated light fields include speckle-structured light fields [31,32,33,34], multi-line-structured light fields [35,36], and binary-structured light fields [37,38,39], while common phase-modulated light fields include single- and multi-frequency sinusoidal gratings [40,41,42,43]. The Gray code method belongs to the category of intensity-modulated light fields. It is a three-dimensional imaging method with good robustness and noise resistance, achieving high precision in object reconstruction. Therefore, in this study, the structured light three-dimensional imaging method based on the Gray code method is employed to reconstruct the physical model of the slope. The reconstruction steps are illustrated in Figure 2.

The Gray Code is a specific sequence of binary number encoding that mandates any adjacent Gray Codes to differ by only one binary digit, and the Gray Codes corresponding to the minimum and maximum numbers differ by only one digit as well. Therefore, the Gray Code is a coding method with minimal errors. During the projection phase, it is essential to design the Gray Code patterns required for the experiment.

### 2.1. Design of Gray Code Coding Pattern

To uniquely encode each pixel in the image using Gray Code values, it is necessary to establish the pixel coordinate system on a two-dimensional plane. Encoding is executed independently in both the horizontal and vertical directions of the image, with the stipulation that the resolution of the Gray Code projection pattern is not inferior to that of the projector. Thus, presuming the projector’s resolution is denoted as L×W pixels, the horizontal stripe level l and vertical stripe level w of the designated Gray Code pattern must adhere to Equation (1):(1)$$\left\{ \begin{array}{l} l=log_{2}^{L}\\ w=log_{2}^{W} \end{array}\right.$$

After devising the Gray code striped levels, for each level of the striped encoding pattern, a corresponding Gray code pattern is designed with the same stripe level, as well as with inverted positions of black and white stripes. This facilitates a more precise computation of grayscale values within the regions covered by the Gray Code patterns on the object under measurement. Due to potential variations in the object’s illumination angles, leading to shadowed areas in the captured images, the early identification of these shadow regions during the decoding process can markedly enhance decoding efficiency. Consequently, two supplementary patterns, one black and one white, are projected onto the object to mitigate the impact of shadow regions [47]. Given that the projector’s resolution is denoted by L×W pixels, it becomes imperative to devise (l+w)×2+2 Gray Code patterns.

### 2.2. Gray Code Decoding

Decoding the Gray Code is the process of obtaining the decimal code value corresponding to each pixel’s Gray Code value. The decoding calculations are shown in Equation (2) [48]:(2)$$V\left(x,y\right)=\sum_{i=1}^{m}GC_{i}\left(x,y\right)\times 2^{\left(m-i\right)}$$
where V represents the decimal code value of the Gray Code; m represents the total number of Gray Code patterns projected; and $GC_{i}$ represents the binarized value of the i-th Gray code image obtained by the camera.

### 2.3. Calibration of Structured Light System

To achieve precise 3D object reconstruction, the geometric relationships and internal parameters of the structured light system play a crucial role. This necessitates calibrating the structured light system before undertaking 3D object reconstruction to establish these transformation relationships. Figure 3 illustrates the structured light system’s calibration process, comprising two components: camera calibration and projector calibration.

In structured light systems, the spatial relationship between the camera and projector is crucial for determining the precision and quality of 3D measurements. Increasing the distance between these components typically results in a reduction in depth resolution, impacting the sharpness and accuracy of the 3D data. However, this expanded distance also allows for a wider field of view, enabling the capture of larger areas in a single scan. This benefit, though, is often counterbalanced by a noticeable decrease in the intricacy and detail of the captured data, as the spread of the light pattern over a larger area reduces its distinctiveness and the system’s ability to discern fine details [49,50]. During the photography process in this study, the camera was positioned approximately 1.5 m from the slope surface, while the projector was placed approximately 1.2 m away from it. The distance between the projector and the camera was approximately 0.3 m.

#### 2.3.1. Calibration of the Camera

In an ideal scenario, a pinhole camera adheres to the principles of linear perspectives, as illustrated in Figure 4 [51]. The transformation relationship for a point P on the object from coordinates $\left(X^{W},Y^{W},Z^{W}\right)$ in the world coordinate system to the projected coordinates (u,v) in the pixel plane coordinate system is defined by Equation (3) [52]:(3)$$Z^{C}\left[\begin{array}{c} u\\ v\\ 1 \end{array}\right]=\left[\begin{array}{cccc} f_{x} & 0 & u_{0} & 0\\ 0 & f_{y} & v_{0} & 0\\ 0 & 0 & 1 & 0 \end{array}\right]\left[\begin{array}{cc} R & T\\ \vec{0} & 1 \end{array}\right]\left[\begin{array}{c} X^{W}\\ Y^{W}\\ Z^{W}\\ 1 \end{array}\right]=KW\left[\begin{array}{c} X^{W}\\ Y^{W}\\ Z^{W}\\ 1 \end{array}\right]$$
where $Z^{C}$ represents the scale factor in the linear mapping from three-dimensional to two-dimensional space. $f_{x}$ and $f_{y}$ represent the normalized focal lengths along the u and v axes, respectively. $u_{0}$ and $v_{0}$ are the pixel coordinates of the camera optical center. The rotation matrix is denoted as R, and the translation matrix as T. Parameters K and W represent the intrinsic and extrinsic parameters, respectively.

In this study, the calibration object employed is a chessboard grid calibration board with known dimensions. The world coordinate system is established by taking the plane where the calibration board is located as the $O-X^{W}Y^{W}$ plane. All corner points on the chessboard grid are situated on this $O-X^{W}Y^{W}$ plane, rendering their coordinates in the Z direction as zero [53]. As the dimensions of each grid on the chessboard grid are known, it enables the computation of the world coordinates for each corner point. The calibration board images are captured, and corner detection algorithms are applied to determine the pixel coordinates of each corner point in the image. Figure 5 displays the detection results of these corner points. A system of calibration equations can be established as shown in Equation (4):
(4)$$\left[\begin{array}{c} u\\ v\\ 1 \end{array}\right]=\frac{1}{Z^{C}}\left[\begin{array}{ccc} f_{x} & 0 & u_{0}\\ 0 & f_{y} & v_{0}\\ 0 & 0 & 1 \end{array}\right]\left[\begin{array}{ccc} r_{1} & r_{2} & T \end{array}\right]\left[\begin{array}{c} X^{W}\\ Y^{W}\\ 1 \end{array}\right]=H\left[\begin{array}{c} X^{W}\\ Y^{W}\\ 1 \end{array}\right]$$
where $r_{1}$ and $r_{2}$ represent the first and second column vectors of the rotation matrix *R*, respectively; and H denotes a 3 × 3 homography matrix.

The pixel coordinates and world coordinates of the multiple sets of acquired corner points are substituted into the calibration equation set (4). The homography matrix H is then computed using the Singular Value Decomposition (SVD) method. Utilizing the acquired homography matrix H, an overdetermined system of equations is established. To solve the system of equations, images of the chessboard calibration pattern need to be captured from at least three distinct angles. The Levenberg–Marquardt (LM) algorithm is employed to solve the overdetermined system of equations, thereby obtaining the intrinsic and extrinsic parameter matrices of the camera.

#### 2.3.2. Calibration of the Projector

The camera serves as an image capture device, while the projector functions as an image output device. Presently, the prevailing calibration method for projectors treats the projector as a camera model with an opposing working principle [54,55]. In this model, the projector adheres to the same imaging principles as a camera. In this study, a decoding method based on this model is employed to calibrate the projector through the projection of Gray codes. The projection process is depicted in Figure 6, where the Gray code patterns are sequentially projected onto a standard black-and-white chessboard grid.

As illustrated in Figure 7, assume that the coordinates of a corner point P on the chessboard grid calibration plate are $\left(X^{W},Y^{W}\right)$. Its corresponding projection on the camera imaging plane is point $P_{c}$, with pixel coordinates $\left(u_{c},v_{c}\right)$. The projection of point P on the projector’s projection plane is point $P_{p}$, with pixel coordinates $\left(u_{p},v_{p}\right)$. As described in Section 2.1, Gray code values are uniquely assigned to each pixel in the projected image. By analyzing the position of point $P_{c}$ within the black and white stripes of a series of grayscale-encoded patterns, its grayscale code values $\left(C_{u},C_{v}\right)$ can be determined. In this context, $C_{u}$ represents the Gray code value along the *u*-axis (horizontal direction), while *C_v_* denotes the Gray code value along the v-axis (vertical direction). Decoding the Gray code value of point $P_{c}$ allows us to obtain the pixel coordinates $\left(u_{p},v_{p}\right)$ of point $P_{p}$. Utilizing this method, the coordinates of the chessboard grid corners in the projector’s pixel coordinate system can be determined [56]. This subsequently enables the calibration of the projector through the application of camera calibration techniques.

## 3. Three-Dimensional Modeling and Analysis for Slope Models

In this experiment, a structured light measurement system for the slope modeling consisted of a physical slope model, a camera, a projector, and post-processing software. As illustrated in Figure 8, the collected soil was applied to a rigid platform to create the physical slope model. The platform, constructed from stainless steel, was divided into two sections: the slope section (85 cm × 100 cm × 10 cm) and the base section (40 cm × 100 cm × 10 cm). The white trapezoidal area was the actual modeling area for this experiment. Inclination angles of the slope section were controlled by adjusting the height of the bolts on both sides. The camera and projector were connected to a tripod. The tripod facilitated convenient adjustment of the camera and projector’s height and direction to fulfill the measurement requirements of the structured light system. Table 1 and Table 2 present detailed parameters of the equipment used in this experiment. Figure 9 illustrates the workflow of the slope model test.

### 3.1. Accuracy Verification Experiment

This section reconstructs the 80 mm radius gypsum sphere depicted in Figure 10a in three dimensions to verify the reconstruction accuracy of the structured light system in this paper. The gypsum sphere serves as an ideal object for three-dimensional reconstruction. Throughout the experimental procedure, the sphere was placed approximately 1.5 m in front of the camera, and the obtained point cloud of the sphere’s surface is presented in Figure 10b. This study employed CloudCompare (v2.12.4) software to perform spherical fitting on the acquired point cloud of the sphere, and the root-mean-square error (RMSE) of the point cloud to the fitted sphere distance was subsequently determined. Concurrently, the radius of the fitted sphere was computed and subsequently compared with the actual size of the gypsum sphere, as shown in Table 3.

The RMSE between the point cloud and the fitted spherical surface in the experimental results is 0.22 mm. This outcome demonstrates that, under relatively ideal reconstruction conditions, the structured light system described in this study is capable of achieving sub-millimeter accuracy.

To ascertain the reconstruction accuracy of the structured light system for the slope materials discussed in this article, a three-dimensional scanning of the slope model was carried out using the Free Scan Combo laser scanner produced by Xianlin (Hangzhou, China) Company. This scanner is capable of capturing three-dimensional data with a precision of 0.02 mm. Table 4 details the comprehensive specifications of this equipment. Given the scanner’s ability to scan the slope model from various angles and positions, it effectively mitigates the impact of light shadowing in the 3D model reconstruction process. The reconstruction model obtained from this laser scanner was used as a benchmark to compare with the models reconstructed by the structured light system described in this study. A comparison between two models was conducted by calculating the distance from cloud to cloud between the models.

In this study, CloudCompare software was used to compute cloud-to-cloud distances. Two different algorithms are used in this software: Closest-to-Closest (C2C) distance and Multiscale Model-to-Model Comparison (M3C2). The C2C algorithm identifies the closest points in the reference cloud and calculates the Euclidean distance, allowing for a quicker completion of point cloud comparison. Nevertheless, this algorithm is sensitive to the roughness and presence of outliers in the point cloud, thereby limiting its effectiveness. Conversely, the M3C2 algorithm utilizes the local surface normals of each point’s neighborhood to calculate point cloud variations, effectively mitigating the impact of point cloud roughness and outliers on the comparison results. This algorithm enables the direct detection of changes in complex terrain on the point cloud model.

The slope model in this study was constructed using soil samples collected from the field. The surface of the slope model is relatively rough. Consequently, the M3C2 algorithm was employed to calculate the distance between the two reconstructed point cloud models. The distribution of distance errors for the reconstruction model of the structured light system is presented in Table 5 and Figure 11. The histogram of M3C2 distances exhibits a normal distribution, with 99% of the points having an M3C2 distance of less than 3.61 mm. The reconstruction accuracy of the structured light system discussed in this paper was assessed using the RMSE of the M3C2 distance errors. The RMSE calculated from the M3C2 distance was found to be 1.08 mm.

Figure 12 illustrates the distance errors between the slope model obtained using a laser scanner and the slope model acquired through the structured light system discussed in this paper. Since these two point cloud models are not in the same coordinate system, registration is necessary before any comparison. Coarse registration is required prior to precise alignment. As depicted in Figure 12, two pieces of gravel of approximately 25 mm in diameter were placed on the surface of the slope model, forming two notable protrusions. This gravel was used as feature objects for the coarse registration of the two point cloud models. The obstruction of light by the gravel prevented the structured light from projecting onto their backside and the adjacent areas. This led to missing point cloud data in these regions in the model reconstructed by the structured light system, while the laser scanner model included these data. Consequently, the lack of point cloud data in the gravel regions of the model reconstructed by the structured light system led to the most significant errors. Therefore, the largest positive and negative distance errors occur in the two gravel regions, which are indicated by the deep red areas in Figure 12.

The results of two distinct experiments indicate that different materials impact the reconstruction accuracy of the structured light system discussed in this paper. However, the structured light system described herein is still capable of achieving an approximate accuracy of 1 mm in the three-dimensional reconstruction of static slope models.

### 3.2. Acquisition of Parameters for Structured Light System

Before initiating image acquisition of the slope model, calibration of the structured light system is necessary. In this study, a projector resolution of 1920 × 1080 pixels is employed. Following the design method outlined in Section 2.1, the Gray code patterns are configured with 11 levels for both horizontal and vertical stripes. In total, 46 Gray code patterns are designed to fulfill the calibration requirements. Calibration results for the structured light system obtained following the calibration steps outlined in Section 2.3 are presented in Table 6.

### 3.3. Simulation and Deformation Measurement of Slope Model Landslide Hazards

After the calibration of the structured light system, the collection of slope model images was initiated. The basic test conditions for the physical model test in this paper adhere to the standard protocol established by Fang et al. [57], as illustrated in Appendix A Figure A1. In this study, the slope angle was systematically adjusted to 31°, 37°, 43°, and 49° to simulate landslide phenomena in the slope model. After stabilization of the landslide phenomenon on slopes with various inclinations, the slope angle was readjusted to the initial 31° to maintain consistency in the captured image area on the slope surface. Figure 13 illustrates the 3D models of the slope following landslide occurrences at various inclinations.

A comparative analysis of 3D slope models created at various slope angles offers a visual comprehension of the extent of slope variations, aiding in the monitoring of areas prone to disasters.

The M3C2 algorithm was utilized in this paper to compute the point cloud variations of neighboring slope models, as illustrated in Figure 14.

In Figure 14a, as the slope gradient increases, the elevation of the slope surface exhibits undulations, which are highlighted using a gradient color scheme representing positive and negative values. Between 31° and 37°, the slope surface undergoes a minor descent, resulting in an overall height variation of approximately 2 mm. At the summit of the slope, a small red region is observed, indicating a protrusion in the slope surface. This protrusion is caused by the sliding and accumulation of large soil particles from the area above the photographed region.

From 37° to 43°, the slope surface experiences a substantial landslide. At the slope base, significant soil accumulation results from the sliding in the upper blue area, as depicted in the red zone in Figure 14b. This portion of the slope surface bulges, with a maximum height difference of 30 mm.

From 43° to 49°, with the increasing slope gradient, a large-scale destructive landslide manifests on the slope surface, as illustrated in Figure 14c. The soil in the upper blue region of the slope has experienced considerable slippage, resulting in a maximum height difference of 52 mm. At the base of the slope, a substantial accumulation of soil has transpired due to preceding landslides. Consequently, the soil sliding in the upper section of the slope surface is impeded by the uplifted part of the slope toe, leading to its retention in the middle section and the formation of a red bulging area.

The acquired 3D point cloud data allow us to obtain cross-sections of any part of the slope model. Figure 15 depicts the cross-sectional plane of four slope models with different slopes at the red line in Figure 8. The figure allows for the observation of the general trend of landslides. Slopes ranging from 31° to 37° display minimal deformation, with overlapping slope curves. The 43° slope demonstrates an overall downward shift in comparison to the 37° slope, with the maximum height difference occurring at the slope toe. The 49° slope displays the most extensive landslide, characterized by the most pronounced trend. These findings align with the observations made at the slope model test site.

### 3.4. Comparison of the Landslide Prediction Method of Structured Light with the Landslide Prediction Method of the Laser

Both the structured light landslide prediction method and the laser landslide prediction method are based on analyzing surface changes of slopes to determine whether a specific area on the slope has a propensity for landslides. In this section, three-dimensional reconstructions of slope models with 37° and 43° inclination angles were performed using the structured light system discussed in this paper and the Free Scan Combo model laser scanner. The M3C2 algorithm was applied to calculate the variations between the 37° and 43° point cloud models reconstructed by the structured light system. The same algorithm was also used to calculate the variations between the 37° and 43° point cloud models reconstructed by the laser scanner. The result of point cloud changes is depicted in Figure 16. Due to the impact of gravel markers on the analysis, the point cloud changes in the laser-reconstructed models were calculated both with and without the gravel marker data, as shown in Figure 16b,c. In Figure 16a, the point cloud model reconstructed by the structured light system is missing some data due to the obstruction of light by the gravel markers. For Figure 16a,c, which both exclude gravel marker data, the slope surface changes in the models are essentially consistent, and the maximum subsidence and maximum uplift distance errors are less than 1 mm. In Figure 16a,b, although the slope surface changes in the models are generally consistent, there is a deviation in the maximum settlement distance attributable to the influence of the gravel data.

The experimental results indicate that the slope changes in the models obtained using the structured light landslide prediction method outlined in this paper are largely consistent with those acquired through the laser landslide prediction method. The laser scanning method, which captures three-dimensional reconstruction models with point cloud data from a wider range of perspectives, allows the laser landslide prediction method to reveal more detailed variations in the slope surface. Acquiring slope surface changes in the models mentioned in this paper takes about 15 min using the laser landslide prediction method, while the structured light landslide prediction method requires only 8 min. Therefore, in terms of prediction time, the structured light landslide prediction method possesses a certain advantage.

## 4. Discussion and Conclusions

In the experiments conducted for this study, the three-dimensional modeling of the slope model is performed following the cessation of natural landslides at various gradients, when the shape of the slope surface is stabilized and ceases to change. This paper focuses on the static measurement of the slope model before and after deformation. The Gray code method employed in this research is characterized by its simplicity and robustness, rendering it effective for the three-dimensional reconstruction of static slope surfaces. However, due to the necessity of projecting numerous patterns for the unique encoding of each pixel in the image, the Gray code method is less suitable for real-time, high-speed 3D modeling. Several methods now exist for achieving dynamic 3D reconstruction with structured light technology by altering the encoding patterns of projected structured light [58,59]. In future tests of slope models, the structured light system discussed in this paper can be integrated with existing dynamic modeling technologies to conduct tests on dynamic landslides.

The structured light 3D reconstruction method presented in this paper is primarily applied in the domain of indoor slope model monitoring. In an indoor environment, controlling environmental factors is feasible to mitigate their impact on the efficacy of structured light reconstruction. When employing structured light methods outdoors, it is essential to consider the influence of external light sources on the structured light. Existing outdoor structured light measurement approaches include using invisible light (infrared) as a projection source to diminish the effects of outdoor lighting, and adopting laser projection to enhance the contrast of the structured light patterns, among others [60,61]. In subsequent tests, the light source used in the structured light system of this study can be altered, enabling applications in scenarios with higher illumination.

In this study, a three-dimensional-structured-light-system-based method is presented to measure the surface of slope models with different slopes, which can obtain the deformation from the three-dimensional point cloud model of the slope. The method based on the structured light system in this paper can achieve sub-millimeter accuracy in 3D reconstruction. A color-mapped deformation map derived from the slope point cloud model is established. Based on the regions corresponding to different colors in the map, the deformation of the slope models can be obtained. This can help researchers identify the positions and severity of disasters such as collapse and uplift of the slope through the model tests. Another outstanding advantage of this method is its low cost with convenient operation. The cost of the structured light system in this experiment consists of two parts: (a) hardware including projectors and cameras and (b) processing software. With the development of digital technology, low-cost digital products have become widely popular. The projectors and cameras required for this experiment only need to meet the requirements for projection and shooting, so inexpensive second-hand smartphones and simple projection devices can be purchased to form the structured light system used in this paper. The point cloud processing software CloudCompare used in this experiment is free. In general, a relatively low cost is sufficient to complete the construction of the structured light system in this experiment.

## Figures and Tables

**Figure 1 sensors-24-00794-f001:**
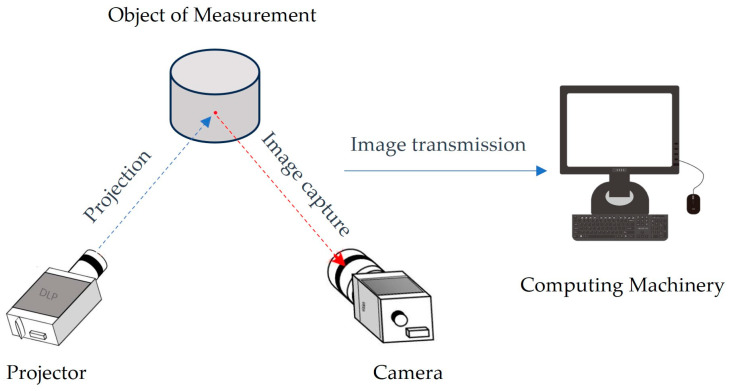
The schematic diagram of the structured light three-dimensional imaging system.

**Figure 2 sensors-24-00794-f002:**
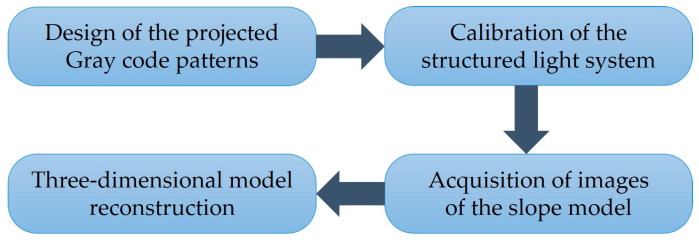
Structured light 3D reconstruction process based on Gray code.

**Figure 3 sensors-24-00794-f003:**
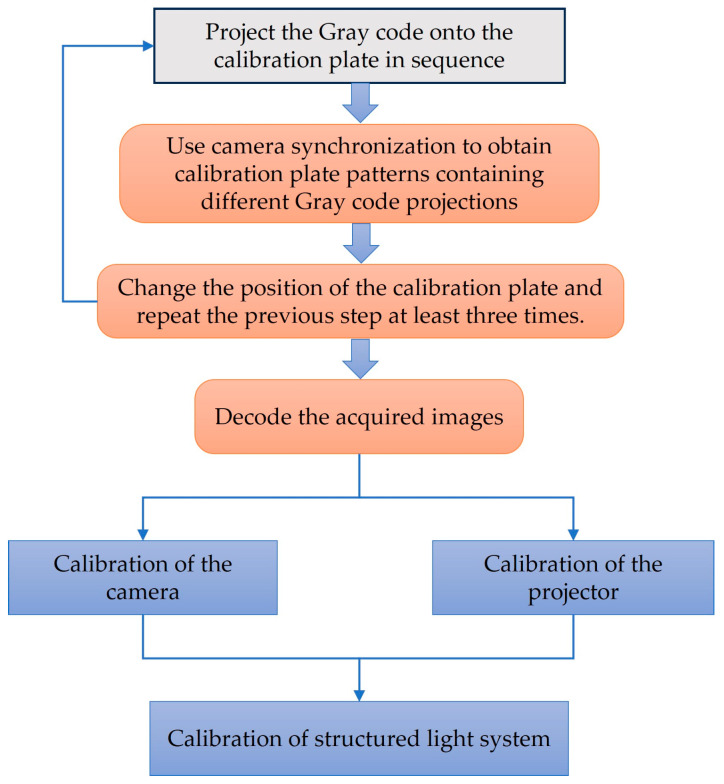
Calibration process of structured light systems.

**Figure 4 sensors-24-00794-f004:**
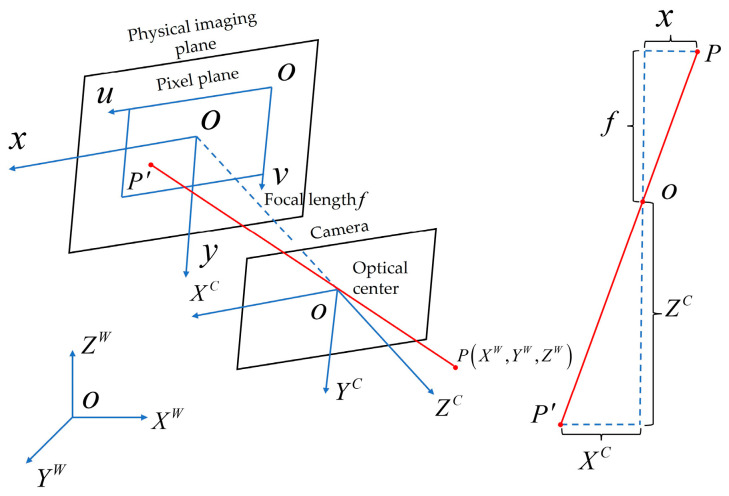
Pinhole camera model. ($O-X^{W}Y^{W}Z^{W}$ represents the world coordinate system; $O-X^{C}Y^{C}Z^{C}$ denotes the camera coordinate system; O−xy is defined as the image coordinate system; O−uv signifies the pixel coordinate system. The red line illustrates the correspondence between point P and point $P^{\prime }$).

**Figure 5 sensors-24-00794-f005:**
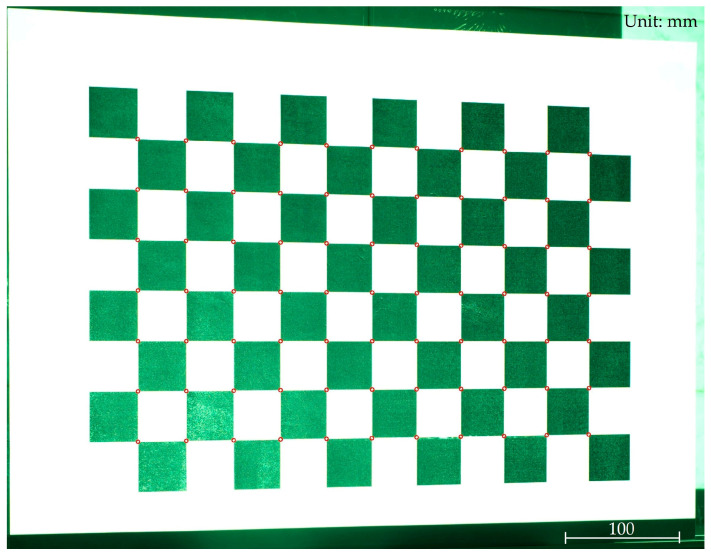
Recognition results of checkerboard grid corner points. (The red circles denote the positions of the corner points that have been identified).

**Figure 6 sensors-24-00794-f006:**
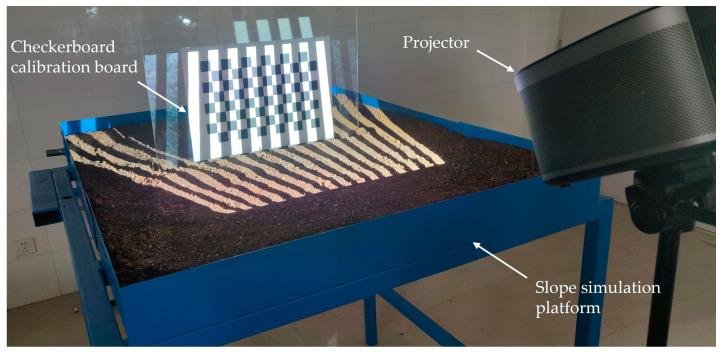
The projection stage in the calibration process of structured light system.

**Figure 7 sensors-24-00794-f007:**
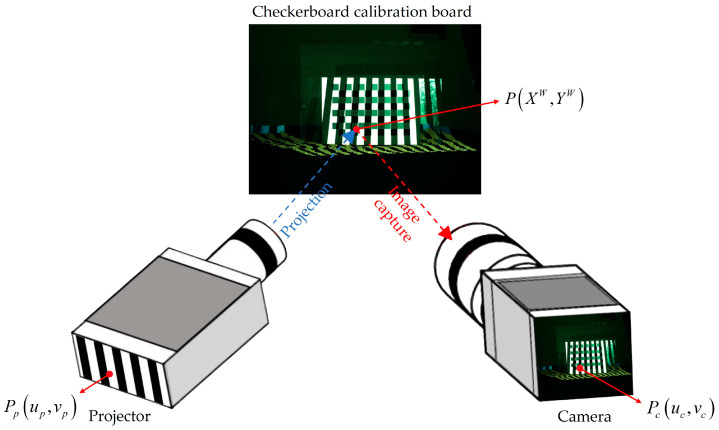
The corresponding positions of the chessboard grid corners on the camera imaging plane and the projector’s projection plane. (The solid line arrows indicate the positions of the corresponding points).

**Figure 8 sensors-24-00794-f008:**
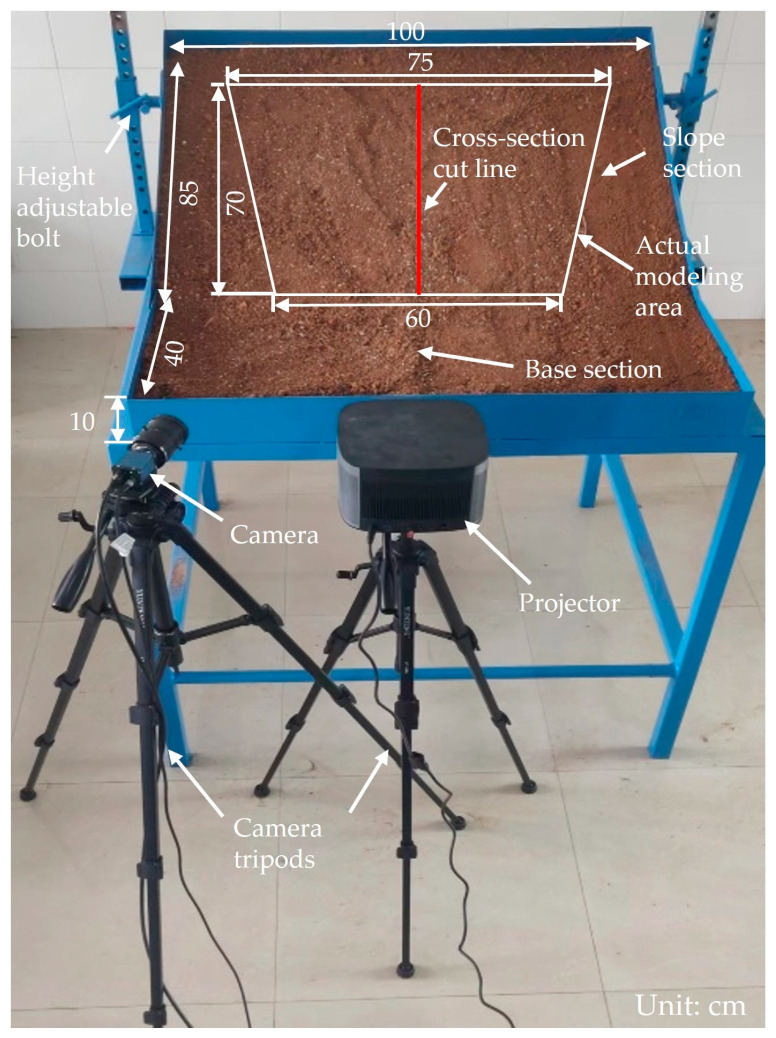
Structured light measurement system for slope modeling.

**Figure 9 sensors-24-00794-f009:**
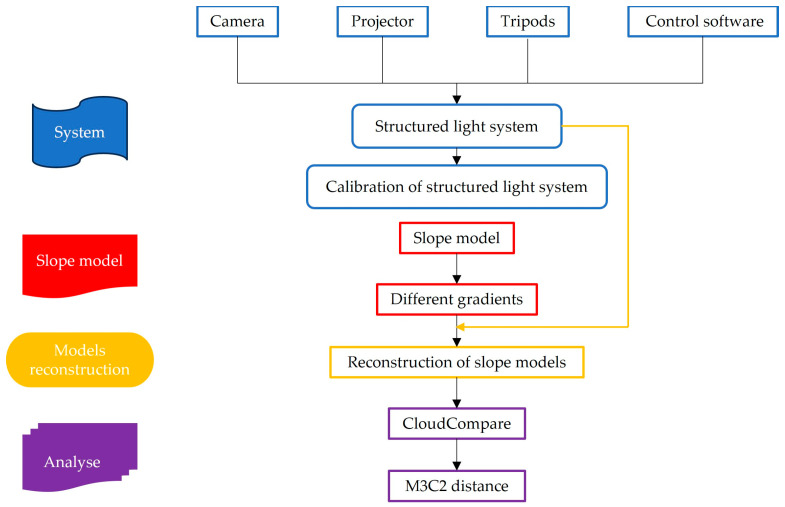
Workflow of the slope model test.

**Figure 10 sensors-24-00794-f010:**
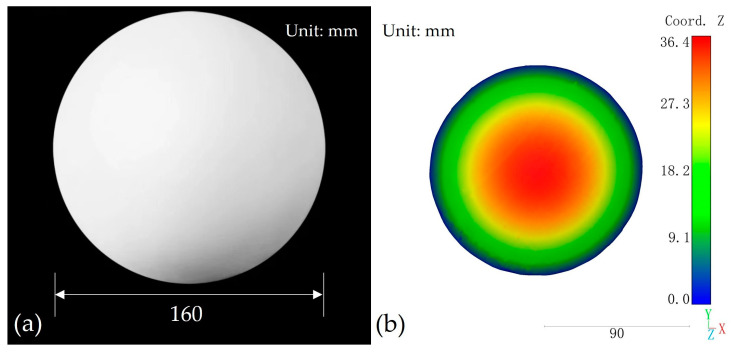
(**a**) Gypsum spherical physical photograph; (**b**) 3D model of gypsum spherical surface.

**Figure 11 sensors-24-00794-f011:**
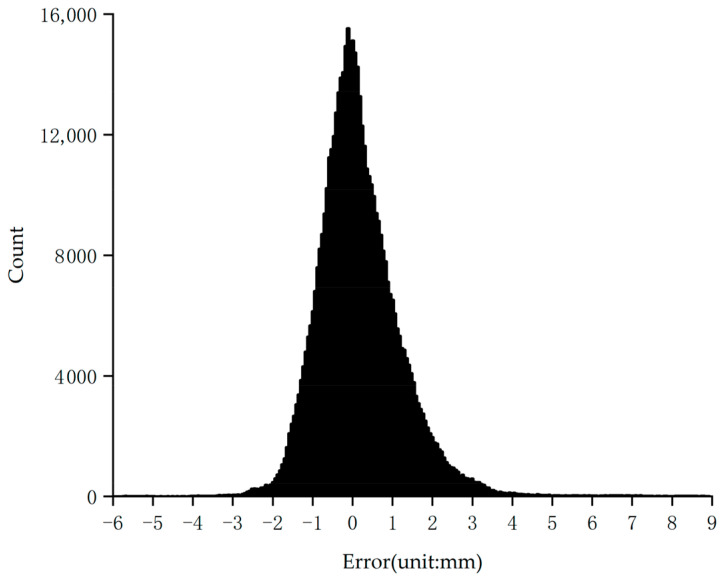
The histogram depicting the distribution of distance errors between the slope model reconstructed by the structured light system and the slope model reconstructed by laser scanning was calculated using the M3C2 algorithm.

**Figure 12 sensors-24-00794-f012:**
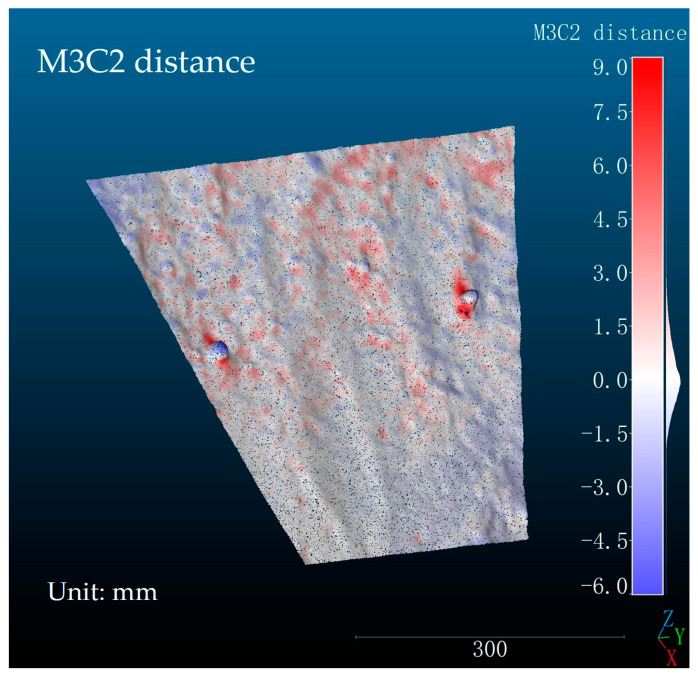
The distribution of distance errors in the slope model reconstructed by the structured light system.

**Figure 13 sensors-24-00794-f013:**
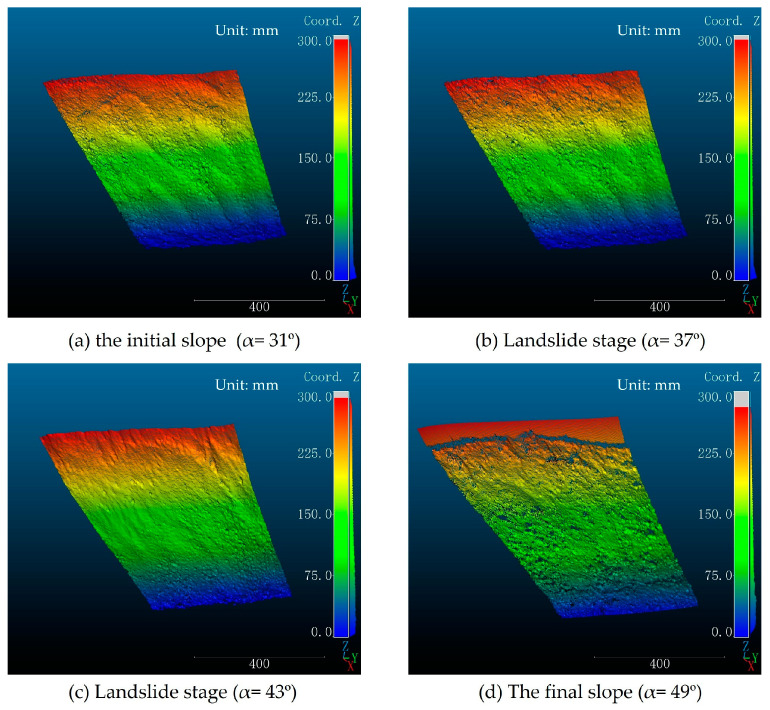
Three-dimensional models of slope failure at different slope angles.

**Figure 14 sensors-24-00794-f014:**
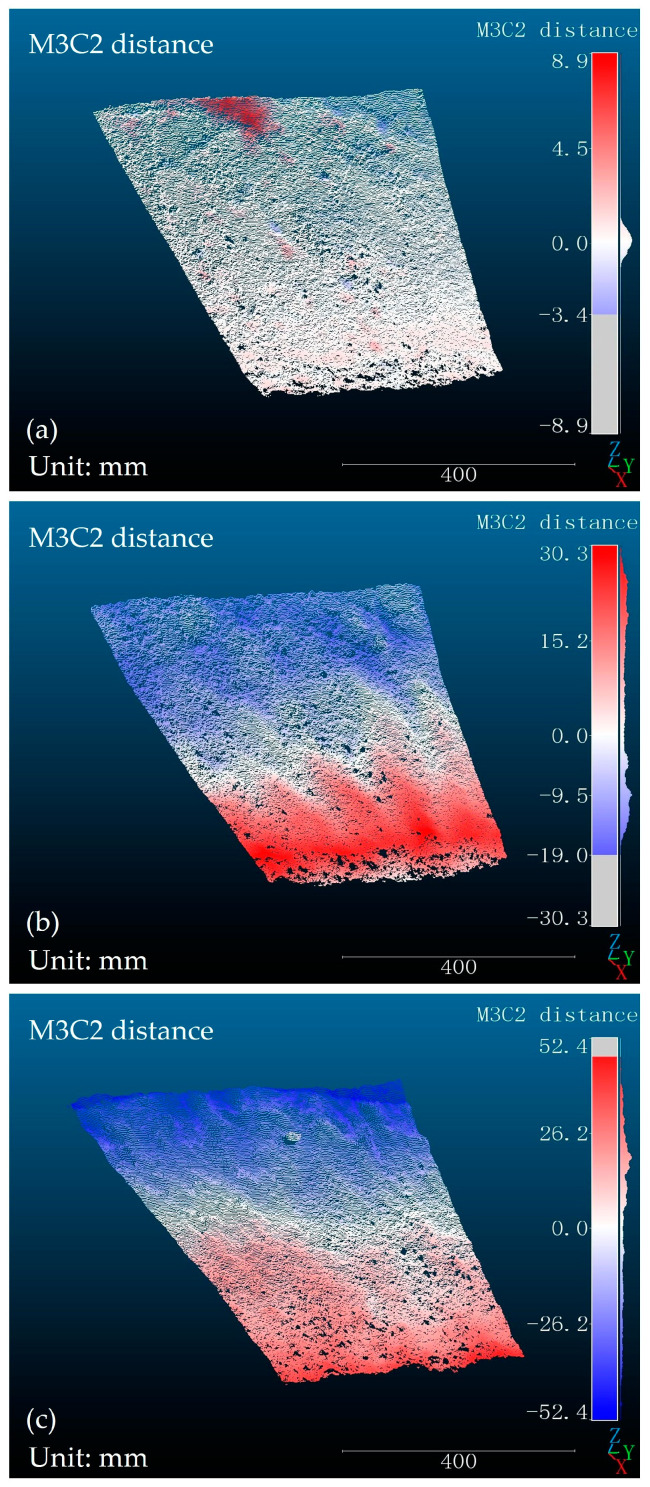
The point cloud variations of adjacent slope models: (**a**) Slope change (α = 31°~37°); (**b**) Slope change (α = 37°~43°); (**c**) Slope change (α = 43°~49°).

**Figure 15 sensors-24-00794-f015:**
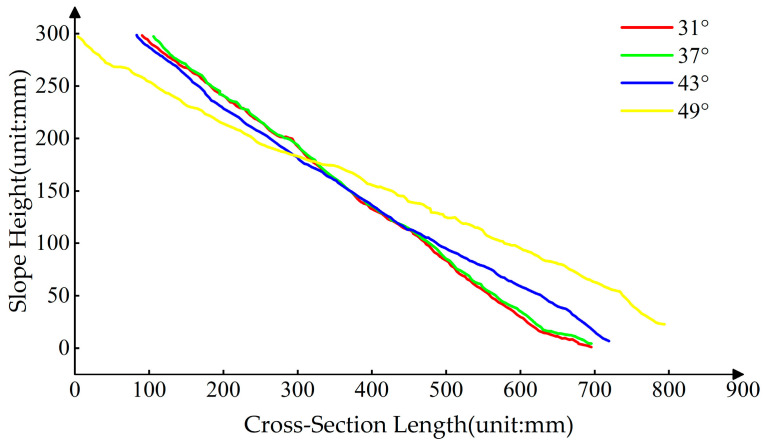
Cross-Section of slope models at different slopes.

**Figure 16 sensors-24-00794-f016:**
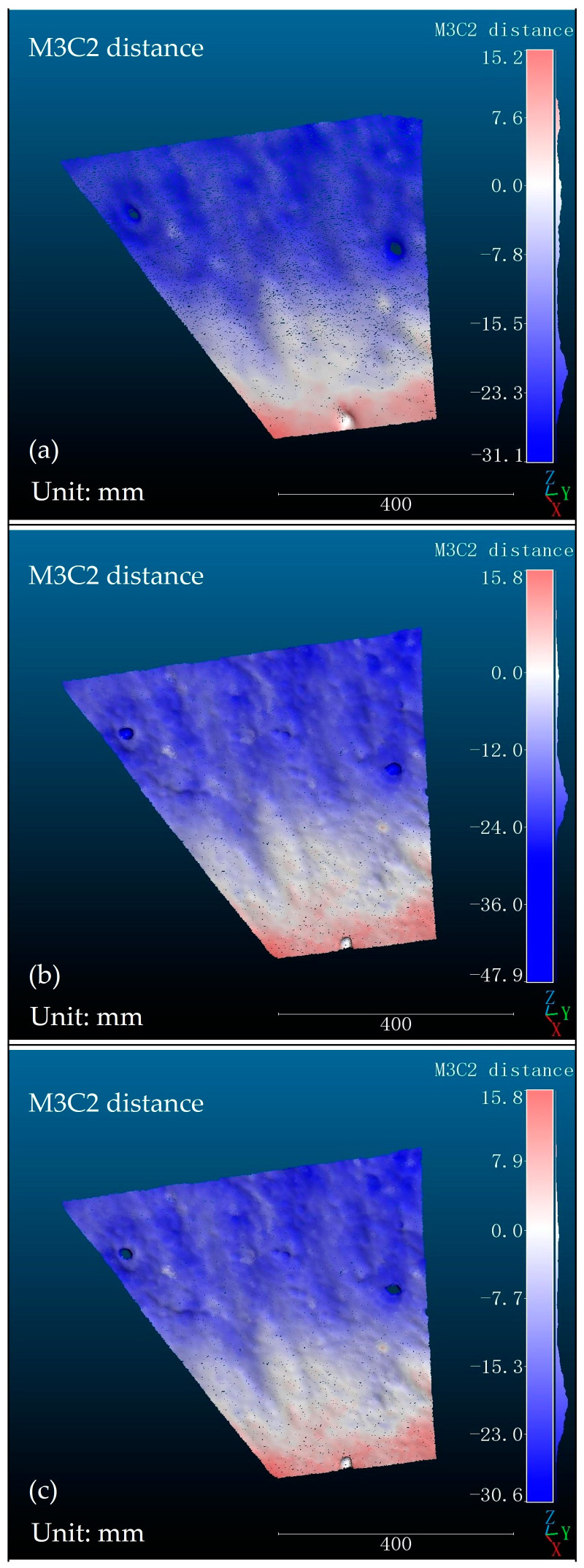
Comparative analysis of slope surface changes in models using structured light and laser landslide prediction methods: (**a**) The landslide prediction method of structured light; (**b**) The landslide prediction method of the laser (Includes gravel marker data); (**c**) The landslide prediction method of the laser (Remove gravel marker data).

**Table 1 sensors-24-00794-t001:** Specifications of the camera in this study.

Manufacturer	Model	Standard Resolution	Luminance	Contrast
XGIMI (Beijing, China)	H3S	1920 × 1080	2200ANSI lm	1000:1

**Table 2 sensors-24-00794-t002:** Specifications of the projector in this study.

Manufacturer	Model	Sensor Type	Sensor Size	Image Format	Focal Length	Aperture
DAHENG (Beijing, China)	HF2514V-2	CCD	1.1 in.	4096 × 3000	25 mm	f/1.4

**Table 3 sensors-24-00794-t003:** The error analysis of gypsum sphere’s 3D reconstruction result (unit: mm).

Type	Evaluation Metrics	Calculation Result
Sphere (r = 80)	Fit Radius	80.13
	Absolute Error	0.13
	RMSE	0.22

**Table 4 sensors-24-00794-t004:** Specifications of the laser scanner in this study.

Manufacturer	Model	Dimensions	Scanning Area	Accuracy	Scan Speed	Working Distance
XIANLIN (Hangzhou, China)	FreeScan Combo	193 × 63 × 53 mm	1000 × 800 mm	0.02 mm	3,500,000 scan/s	300 mm

**Table 5 sensors-24-00794-t005:** Distance error of the structured light system reconstruction model in the test.

Type	99%	RMSE
M3C2	3.61 mm	1.08

**Table 6 sensors-24-00794-t006:** Calibration parameters of the structured light system in this study.

Hardware Type	Parameter Type	Parametric Expression	Calibration Result
Camera	Internal reference matrix	$A^{c}=\left[\begin{array}{ccc} {f_{x}}^{c} & 0 & {u_{0}}^{c}\\ 0 & {f_{y}}^{c} & {v_{0}}^{c}\\ 0 & 0 & 1 \end{array}\right]$	$\left[\begin{array}{ccc} 7623.55 & 0 & 2064.47\\ 0 & 7592.81 & 1390.26\\ 0 & 0 & 1 \end{array}\right]$
	Aberration coefficient	$\left[{k_{1}}^{c},{k_{2}}^{c},{p_{1}}^{c},{p_{2}}^{c},{k_{3}}^{c}\right]$	[−0.249,23.022,−0.002,0.003,0]
Projector	Internal reference matrix	$A^{p}=\left[\begin{array}{ccc} {f_{x}}^{p} & 0 & {u_{0}}^{p}\\ 0 & {f_{y}}^{p} & {v_{0}}^{p}\\ 0 & 0 & 1 \end{array}\right]$	$\left[\begin{array}{ccc} 586.27 & 0 & 495.58\\ 0 & 587.26 & 400.81\\ 0 & 0 & 1 \end{array}\right]$
	Aberration coefficient	$\left[{k_{1}}^{p},{k_{2}}^{p},{p_{1}}^{p},{p_{2}}^{p},{k_{3}}^{p}\right]$	[0.073,−0.148,0.006,−0.001,0]
Structured Light System	Rotation matrix	$R=\left[\begin{array}{ccc} r_{11} & r_{22} & r_{33}\\ r_{21} & r_{22} & r_{23}\\ r_{31} & r_{32} & r_{33} \end{array}\right]$	$\left[\begin{array}{ccc} 0.957 & 0.014 & 0.288\\ 0.110 & 0.906 & -0.410\\ -0.267 & 0.423 & 0.866 \end{array}\right]$
	Translation vector	$T={\left[\begin{array}{ccc} t_{1} & t_{2} & t_{3} \end{array}\right]}^{T}$	${\left[\begin{array}{ccc} -576.746 & 428.688 & -635.508 \end{array}\right]}^{T}$

## Data Availability

The data used to support the findings of this study are available from the corresponding author upon request.

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
