# Peer review of "Three-Dimensional Reconstruction and Deformation Identification of Slope Models Based on Structured Light Method"

_sensors, 2024, doi:10.3390/s24030794_

Round 1

Reviewer 1 Report

Comments and Suggestions for Authors

This manuscript introduces a Structured Light-based 3D reconstruction method for slope models. Overall, the manuscript is well-organized. However, I think some issues should be addressed, as follows:

1. Landslide, i.e, deformation of the slope is dynamic processing, which means the 3D model is changing. However, structured light methods need multiple images with different structured light, e.g., the used gray code. This means the 3D model should be fixed. The authors should further discuss the limitations of your application and try to solve this problem.

2. In section 3.1 Accuracy Verification Experiment, the author uses an 80 mm-radius gypsum sphere to quantitatively test the accuracy of the method. However, the material of the sphere is very different from the slope, which may not accurately illustrate the results. I think the authors should further discuss or show experiments on different materials.

3. Lacks introduction to other photometry methods for accurate three-dimensional modeling that may be used for slope models, such as photometric stereo [1-3].

[1] Fast defect inspection based on data-driven photometric stereo

[2] Estimating high-resolution surface normals via low-resolution photometric stereo images

[3] PS-FCN: A flexible learning framework for photometric stereo

Author Response

January 15 2024

Dear Reviewer:

I am pleased to receive your reviews on our manuscript (Manuscript ID: 2805343). We thank your efforts, which have helped us to improve the overall quality of the paper.

The paper has been revised according to the comments.  All revisions are marked in red. The following is our reply (marked in red) to reviewer’ questions/comments. Introduction has been revised to provide more sufficient background with adding relevant references. The research design and methods are introduced in more detail. The conclusions are revised with adding more results.

We look forward to hearing from you.

Sincerely,

Changxing Zhang

------------------

Changxing Zhang,Ph.D

Associate Professor

Department of Engineering Mechanics, Faculty of Civil Engineering and Mechanics, Kunming University of Science and Technology

Yunnan,650500, P.R.China

Tel: 86-13671380669(O)

Email: zhangcx@kust.edu.cn

Reviewer 1

This manuscript introduces a Structured Light-based 3D reconstruction method for slope models. Overall, the manuscript is well-organized. However, I think some issues should be addressed, as follows:

  1. Landslide, i.e, deformation of the slope is dynamic processing, which means the 3D model is changing. However, structured light methods need multiple images with different structured light, e.g., the used gray code. This means the 3D model should be fixed. The authors should further discuss the limitations of your application and try to solve this problem.

Our replies:

Thank you for your valuable comments. We discussed the limitations of structured light systems in dynamic slope modeling and the directions for improvement in future testing in the discussion and conclusion section of this article. The specific added content is added in discussion and conclusion from line 364 to 376 in the article as follows:

"In the experiments conducted for this study, the three-dimensional modeling of the slope model was performed following the cessation of natural landslides at various gradients, when the shape of the slope surface stabilized and ceased to change. This paper focuses on the static measurement of the slope model before and after deformation. The Gray code method employed in this research is characterized by its simplicity and robustness, rendering it effective for the three-dimensional reconstruction of static slope surfaces. However, due to the necessity of projecting numerous patterns for the unique encoding of each pixel in the image, the Gray code method is less suitable for real-time, high-speed 3D modeling. Several methods now exist for achieving dynamic 3D reconstruction with structured light technology by altering the encoding patterns of projected structured light [58,59]. In future tests of slope models, the structured light system discussed in this paper can be integrated with existing dynamic modeling technologies to conduct tests on dynamic landslides."

Two references are added as follows:

[58] Bell T, Li B, Zhang S. Structured light techniques and applications[J]. Wiley Encyclopedia of Electrical and Electronics Engineering, 1999: 1-24.

[59] Zhang S. High-speed 3D shape measurement with structured light methods: A review[J]. Optics and lasers in engineering, 2018, 106: 119-131.

  1. In section 3.1 Accuracy Verification Experiment, the author uses an 80 mm-radius gypsum sphere to quantitatively test the accuracy of the method. However, the material of the sphere is very different from the slope, which may not accurately illustrate the results. I think the authors should further discuss or show experiments on different materials.

Our replies:

Thank you for your valuable comments. We added an experiment to verify the reconstruction accuracy of the slope material discussed in this article using a structured light system in the accuracy verification section. This experiment involves comparing the model reconstructed from the laser scanner with the model obtained using the structured light system described in this paper. The M3C2 algorithm is used to calculate the distance between the two reconstructed point cloud models. The reconstruction accuracy of the structured light system is evaluated using the root mean square error (RMSE) of the M3C2 distance error. The results of two distinct experiments indicate that different materials impact the reconstruction accuracy of the structured light system discussed in this paper. However, the structured light system described herein is still capable of achieving an approximate accuracy of 1mm in the three-dimensional reconstruction of static slope models. The specific added content is from line 229 to 281 in the article as follows:

" To ascertain the reconstruction accuracy of the structured light system for the slope materials discussed in this article, a three-dimensional scanning of the slope model was carried out using the Free Scan Combo laser scanner produced by Xianlin Company. This scanner is capable of capturing three-dimensional data with a precision of 0.02 mm. Table 4 details the comprehensive specifications of this equipment. Given the scanner's ability to scan the slope model from various angles and positions, it effectively mitigates the impact of light shadowing in the 3D model reconstruction process. The reconstruction model obtained from this laser scanner was used as a benchmark to compare with the models reconstructed by the structured light system described in this study. A comparison between two models is conducted by calculating the distance from cloud to cloud between the models.

In this study, CloudCompare software was used to compute cloud-to-cloud distances. Two different algorithms are used in this software: Closest-to-Closest (C2C) distance and Multiscale Model-to-Model Comparison (M3C2). The C2C algorithm identifies the closest points in the reference cloud and calculates the Euclidean distance, allowing for a quicker completion of point cloud comparison. Nevertheless, this algorithm is sensitive to the roughness and presence of outliers in the point cloud, thereby limiting its effectiveness. Conversely, the M3C2 algorithm utilizes the local surface normals of each point's neighborhood to calculate point cloud variations, effectively mitigating the impact of point cloud roughness and outliers on the comparison results. This algorithm enables the direct detection of changes in complex terrain on the point cloud model.

The slope model in this study was constructed using soil samples collected from the field. The surface of the slope model is relatively rough. Consequently, the M3C2 algorithm was employed to calculate the distance between the two reconstructed point cloud models. The distribution of distance errors for the reconstruction model of the structured light system is presented in Table 5 and Figure 11. The histogram of M3C2 distances exhibits a normal distribution, with 99% of the points having an M3C2 distance of less than 3.61 mm. The reconstruction accuracy of the structured light system discussed in this paper is assessed using the RMSE of the M3C2 distance errors. The RMSE calculated from the M3C2 distance is found to be 1.08 mm.

TABLE 4. Specifications of the laser scanner in this study.

Model

Dimensions

Scanning area

Accuracy

Scan speed

Working distance

XIANLIN

FreeScan Combo

1936353 mm

1000800 mm

0.02 mm

3500000 scan/s

300 mm

TABLE 5. Distance error of the structured light system reconstruction model in the test.

Type

99%

RMSE

M3C2

3.61mm

1.08

FIGURE 11. The histogram depicting the distribution of distance errors between the slope model reconstructed by the structured light system and the slope model reconstructed by laser scanning was calculated using the M3C2 algorithm.

FIGURE 12. The distribution of distance errors in the slope model reconstructed by the structured light system.

Figure 12 illustrates the distance errors between the slope model obtained using a laser scanner and the slope model acquired through the structured light system discussed in this paper. Since these two point cloud models are not in the same coordinate system, registration is necessary before any comparison. Coarse registration is required prior to precise alignment. As depicted in Figure 12, two pieces of gravel of approximately 25mm in diameter were placed on the surface of the slope model, forming two notable protrusions. This gravel was used as feature objects for the coarse registration of the two point cloud models. The obstruction of light by the gravel prevented the structured light from projecting onto their backside and the adjacent areas. This led to missing point cloud data in these regions in the model reconstructed by the structured light system, while the laser scanner model included this data. Consequently, the lack of point cloud data in the gravel regions of the model reconstructed by the structured light system led to the most significant errors. Therefore, the largest positive and negative distance errors occur in the two gravel regions, which are indicated by the deep red areas in Figure 12.

The results of two distinct experiments indicate that different materials impact the reconstruction accuracy of the structured light system discussed in this paper. However, the structured light system described herein is still capable of achieving an approximate accuracy of 1mm in the three-dimensional reconstruction of static slope models."

  1. Lacks introduction to other photometry methods for accurate three-dimensional modeling that may be used for slope models, such as photometric stereo [1-3].

[1] Fast defect inspection based on data-driven photometric stereo

[2] Estimating high-resolution surface normals via low-resolution photometric stereo images

[3] PS-FCN: A flexible learning framework for photometric stereo

Our replies:

Thank you for your valuable comments. We have added an introduction to photometric stereo and cited relevant references in the article. The specific added content is from line 56 to 67 in the article as follows:

"Common non-contact methods for acquiring 3D data include the use of 3D laser scanners [25,26], binocular vision [27,28], photometric stereo [29,30] and structured light [31-43]. 3D laser scanners are expensive and time-consuming, limiting their application in small-scale physical slope model tests. The reconstruction area of binocular vision needs to be within the common field of view of the two cameras, posing a challenge to the specific operations involved in the slope model reconstruction process. Photometric stereo can be applied to objects of various sizes with high resolution, by adjusting the low-cost hardware. However, shiny or reflective surfaces may produce specular highlights, which could distort the images captured and result in inaccuracies in the reconstructed surface models. Though machine learning-based methods can improve this issue [44-46], the approach is too complex."

Five references are added as follows:

[29] Woodham R J. Photometric method for determining surface orientation from multiple images[J]. Optical engineering, 1980, 19(1): 139-144.

[30] Barsky S, Petrou M. The 4-source photometric stereo technique for three-dimensional surfaces in the presence of highlights and shadows[J]. IEEE Transactions on Pattern Analysis and Machine Intelligence, 2003, 25(10): 1239-1252.

[44] Ren M, Wang X, Xiao G, et al. Fast defect inspection based on data-driven photometric stereo[J]. IEEE Transactions on Instrumentation and Measurement, 2018, 68(4): 1148-1156.

[45] Ju Y, Jian M, Wang C, et al. Estimating high-resolution surface normals via low-resolution photometric stereo images[J]. IEEE Transactions on Circuits and Systems for Video Technology, 2023.

[46] Chen G, Han K, Wong K Y K. PS-FCN: A flexible learning framework for photometric stereo[C]//Proceedings of the European conference on computer vision (ECCV). 2018: 3-18.

Reviewer 2 Report

Comments and Suggestions for Authors

In this study, a three-dimensional structured light system-based method is presented to measure the surface of slope models with different slopes, which can obtain the deformation from the three-dimensional point cloud model of the slope.

Landslides are typical natural disasters that profoundly and enduringly influence people's lives. The measurement and monitoring of landslide deformation behavior hold crucial significance in preventing and mitigating losses resulting from landslide disasters.

My main concern is whether the method of 3D reconstruction using structured light is suitable for landslide detection application scenarios. If the application is for outdoor scenes, then the effect of outdoor light on the structured light source has to be considered.

The camera calibration method in section 2.3.1 of the paper should have used Zhang Zhengyou's calibration method, not the method in Equation 3. This is because the z-coordinate of the 3D point is 0. Please describe in detail the camera calibration methodology used in this paper.

In the introduction to the calibration method of the projector in section 2.3.2, the authors need to detail how to solve for the coordinates of the corner points of the calibration plate under the coordinate system of the projector.

In the later experimental section, I see that the authors projected Gray codes both horizontally and vertically. This method is going to produce a very large amount of redundant information. The structured light reconstruction method for Gray codes is to consider the structured light projected by the Gray code as a projection plane. Then the projected rays of the camera and the projection plane of the projector are reconstructed in three dimensions by point-plane intersection. Therefore the structured light reconstruction method in this paper itself needs to be reconsidered.

In the experimental section, more relevant experiments should be added. (1) The effect of the distance between the camera and the projector on the reconstruction results. (2) Comparison of structured light 3D reconstruction results with laser 3D reconstruction results. (3) Comparison of the landslide prediction method of structured light with the landslide prediction method of the laser. (4) Comparison of time performance of different methods.

Author Response

January 15 2024

Dear Reviewer:

I am pleased to receive your reviews on our manuscript (Manuscript ID: 2805343). We thank your efforts, which have helped us to improve the overall quality of the paper.

The paper has been revised according to the comments.  All revisions are marked in red. The following is our reply (marked in red) to reviewer’ questions/comments. Introduction has been revised to provide more sufficient background with adding relevant references. The research design and methods are introduced in more detail. The conclusions are revised with adding more results.

We look forward to hearing from you.

Sincerely,

Changxing Zhang

------------------

Changxing Zhang,Ph.D

Associate Professor

Department of Engineering Mechanics, Faculty of Civil Engineering and Mechanics, Kunming University of Science and Technology

Yunnan,650500, P.R.China

Tel: 86-13671380669(O)

Email: zhangcx@kust.edu.cn

Round 2

Reviewer 1 Report

Comments and Suggestions for Authors

Thank you for the detailed responses. All my concerns are addressed and I have no further issues. I strongly recommend accepting this paper in the current version.

Reviewer 2 Report

Comments and Suggestions for Authors

The paper has been revised according to my comments. I think it can be accepted.